# Concordance between Dash Diet and Hypertension: Results from the Mediators of Atherosclerosis in South Asians Living in America (MASALA) Study

**DOI:** 10.3390/nu15163611

**Published:** 2023-08-17

**Authors:** Bridget Murphy Hussain, Andrea L. Deierlein, Alka M. Kanaya, Sameera A. Talegawkar, Joyce A. O’Connor, Meghana D. Gadgil, Yong Lin, Niyati Parekh

**Affiliations:** 1Public Health Program, Marion Peckham Egan School of Nursing and Health Studies, Fairfield University, Fairfield, CT 06824, USA; bhussain@fairfield.edu; 2Public Health Nutrition, School of Global Public Health, New York University, New York, NY 10012, USA; ald8@nyu.edu (A.L.D.); jao8@nyu.edu (J.A.O.); 3Department of Population Health, New York University Langone Health, New York, NY 10016, USA; 4Division of General Internal Medicine, Department of Medicine, University of California, San Francisco, CA 94115, USA; alka.kanaya@ucsf.edu (A.M.K.); meghana.gadgil@ucsf.edu (M.D.G.); 5Departments of Exercise and Nutrition Sciences and Epidemiology, Milken Institute School of Public Health, The George Washington University, Washington, DC 20052, USA; stalega1@email.gwu.edu; 6Department of Biostatistics and Epidemiology, School of Public Health, Rutgers University, Newark, NJ 07103, USA; linyo@sph.rutgers.edu; 7Rory Meyers College of Nursing, New York University, New York, NY 10012, USA

**Keywords:** South Asian, DASH diet, hypertension

## Abstract

High blood pressure is an important predictor of atherosclerotic cardiovascular disease (ASCVD), particularly among South Asians, who are at higher risk for ASCVD when compared to other population groups. The Dietary Approaches to Stop Hypertension (DASH) dietary pattern is established as the best proven nonpharmacological approach to preventing hypertension in adults. Using data from the Mediators of Atherosclerosis in South Asians Living in America (MASALA) cohort, we calculated a DASH dietary score to examine the association between adherence to the DASH diet and its components, and prevalent and incident hypertension and systolic and diastolic blood pressure, after five years of follow-up. We found that the relative risk ratio (RRR) of incident hypertension was 67% lower among participants in the highest DASH diet score category (aRRR: 0.33; 95% CI: 0.13, 0.82; p_trend_ = 0.02) compared with those in the lowest DASH diet score category in fully adjusted models. These findings are consistent with previous clinical trials and large prospective cohort studies, adding to evidence that supports the diet-disease relationship established between DASH diet and hypertension. This study is the first to examine DASH diet adherence and hypertension among South Asian adults in the U.S.

## 1. Introduction

High blood pressure, or hypertension, increases risk for atherosclerotic cardiovascular disease (ASCVD) [1,2]. As South Asians are nearly three times more likely to develop hypertension early in life compared with non-Hispanic whites [3,4], prevention strategies for reducing overall blood pressure are needed in this growing population subgroup in the United States (U.S.). The American College of Cardiology and American Heart Association identify nonpharmacological interventions for the primary prevention of hypertension and ASCVD, specifically highlighting the Dietary Approaches to Stop Hypertension (DASH) dietary pattern as the best proven healthy diet for the prevention and treatment of hypertension [5]. The DASH pattern emphasizes a diet that is rich in fruits, vegetables, nuts, seeds, legumes, whole grains, and low-fat dairy products, which results in high potassium, magnesium, calcium, antioxidant, and fiber intake, while concurrently reducing processed food intake, including sugar-sweetened beverages, red and processed meats, and sodium [6]. Previous randomized controlled trials have indicated that adherence to the DASH diet over an 8-week period resulted in an estimated 11 mm of mercury (mmHg) reduction in systolic blood pressure among non-Hispanic whites and African Americans with diagnosed hypertension, and 3 mmHg among those with normotension [5,7]. Subsequent cohort studies have sought to study DASH dietary adherence using scoring methods that assess and quantify DASH components in usual dietary intake, beyond experimental settings [8]. Although adherence to a DASH-style diet is associated with lower risk of hypertension within several population cohorts [8], studies do not include or disaggregate the dietary patterns of South Asians in the U.S.

Using data from the Mediators of Atherosclerosis in South Asians Living in America (MASALA) study [9], a community-based prospective cohort study of primarily South Asian immigrants in the U.S., we examined prospective associations of the DASH dietary pattern with measures of blood pressure and hypertension, including incident and prevalent hypertension 2, and overall systolic and diastolic blood pressure measurements. We hypothesized that a greater adherence to the DASH dietary pattern would be associated with lower incident hypertension, and lower systolic and diastolic blood pressures.

## 2. Materials and Methods

### 2.1. Study Population

The MASALA study is an ongoing community-based cohort study of South Asian Americans aged 40–84 years, free from cardiovascular disease at enrollment, who reside in the San Francisco or Chicago metropolitan areas [9]. Exclusion criteria included self-reported diagnosis of a stroke, heart attack, heart failure, transient ischemic attack, or angina; use of nitroglycerin; prevalent atrial fibrillation; history of cardiovascular procedures; active treatment for cancer; impaired cognition; plans to move out of the study region in the five years subsequent to enrollment; life expectancy less than five years due to serious medical condition; and residence in (or on a waiting list for) a nursing home. Additional detail on baseline measures and recruitment have been previously published [9]. The MASALA study protocols were approved by the institutional review boards at Northwestern University (NWU) and University of California San Francisco (UCSF). All participants provided written informed consent. The current analyses were approved by the New York University institutional review board (IRB-FY2021-5009; approved on 2 February 2021).

### 2.2. Analytical Dataset

Participants with data from both Exam 1 (baseline, 2010–2013) and Exam 2 (2015–2018) were eligible for inclusion in the analytical sample (*n* = 746). Ninety-eight percent of participants were born outside the U.S. (*n* = 19 s generation South Asians); therefore, we restricted analysis to only those who immigrated to the U.S. Participants missing diastolic or systolic blood pressure readings at Exam 2 were excluded from analysis (*n* = 1). Participants with missing food frequency questionnaire (FFQ) data, implausible caloric intake (<800 kilocalories/day or >4200 kilocalories/day in men; <500 kilocalories/day or >3500 kilocalories/day in women) [10], or missing components of the DASH diet score at Exam 1 were excluded (*n* = 10). The final analytical sample included 716 participants (Figure 1).

### 2.3. Data Collection

Dietary Data: Dietary intake data were collected during Exam 1 by MASALA study staff trained to use and administer the Study of Health Assessment and Risk in Ethnic (SHARE) FFQ [11]. The 163-item interviewer-administered tool is validated to assess dietary intake of South Asian adults living in North America. Food intake was assessed by average serving sizes (e.g., ounces, tablespoons, cups) and frequency (i.e., intake of items per day, week, month, or year) to determine average daily quantities with total caloric intake (kilocalories/day) calculated by the SHARE FFQ report [11].

Blood Pressure: Blood pressure was measured using an automated monitor (v100 Vital Signs Monitor; GE Healthcare, Fairfield, CT, USA). Participants were seated with their feet flat on the floor and three measurements were taken two minutes apart. The average of the second two readings were used for analysis. Blood pressure was measured using systolic (mmHg) over diastolic (mmHg) readings. Incident and prevalent hypertension were assessed at exam 2 using the NCEP criteria [12] of systolic blood pressure reading ≥130 mmHg and/or diastolic blood pressure reading ≥85 mmHg or use of anti-hypertensive medication.

Sociodemographic and Health Behavior Data: Sociodemographic data were collected at Exam 1. Percent of life lived in the U.S. was based on the participant’s age divided into the number of years the participant has lived in the U.S. Annual family income was collected categorically (response choices < $40,000 per year, $40,000–$75,000 per year, $75,000–$100,000 per year, and >$100,000 per year); income was dichotomized as >$100,000 per year and ≤$100,000 per year. Educational attainment was collected categorically and dichotomized as achieved bachelor’s degree (yes or no). Information on health behaviors including smoking status, alcohol intake, and physical activity were collected at Exam 1 and Exam 2. Smoking status was collected categorically (never, former, current) and dichotomized as ever smoking or never smoking. Alcohol intake was categorized based on average drinks per week and dichotomized as no alcohol or any alcohol intake. Physical activity was assessed using the Typical Week’s Physical Activity Questionnaire [13] and categorized using the American Heart Association’s Life’s Simple Seven criteria for poor (no physical activity), intermediate (1–149 min of moderate activity or 1–74 min of vigorous activity), or ideal (≥150 min of moderate activity or ≥75 min of vigorous activity) physical activity levels [14,15].

Clinical Data: Clinical characteristics were collected at Exam 1. Coronary Artery Calcium (CAC) scores were measured using a cardiac-gated electron-beam computed tomography scanner using non-contrast cardiac computed tomography scans (NWU: Siemens Sensation Cardiac 64 Scanner (Siemens Medical Solutions, Malvern, PA, USA); UCSF: Phillips 16D scanner or a Toshiba MSD Aquilion 64). CAC scores were categorized according to the Agatston guidelines (0, 0–100, 101–399, ≥400) [16]. Fasting blood samples were collected after a 12 h fast. High-density lipoprotein (HDL) cholesterol was measured by enzymatic methods (Quest, San Jose, CA, USA) and low-density lipoprotein (LDL) cholesterol was calculated [9]. Diabetes (yes or no) was determined if a participant reported taking a medication for diabetes, had a fasting plasma glucose ≥ 126 mg/dL, or post-glucose challenge value of ≥200 mg/Dl [9]. Participant height was measured using a stadiometer (without shoes) and weight was measured in regular clothing (without shoes) using a digital scale; these values were used to calculate body mass index (BMI; k/m^2^) [9].

### 2.4. Creation of the DASH Diet Score

Using the SHARE FFQ raw data, we computed the DASH dietary concordance score following the Fung et al. method to quantify adherence [17]. The score focuses on a total of eight components aligned with the DASH diet, including high intake of (1) fruit, (2) vegetables, (3) nuts/legumes, (4) low-fat dairy products, and (5) whole grains; and low intake of (6) sodium, (7) sugar-sweetened beverages, and (8) red/processed meats. Participants were classified into quintiles according to intake of each component, with sodium, sugar-sweetened beverage, and red/processed meat intake reverse scored such that higher intake received a lower score. Appendix A depicts the scoring methodology and criteria. The theoretical DASH score ranges from 8 (low concordance) to 40 (high concordance). The DASH score was divided into three categories of diet score, with cut-off scores consistent with previous DASH diet score analyses in which the lowest score category was ≤20 and the highest score category was ≥29 [17,18].

### 2.5. Statistical Analysis

Descriptive statistics (count and percent for discrete variables; mean and standard deviation (SD) for continuous variables) for sociodemographic characteristics, caloric intake, health behaviors (smoking, alcohol intake, physical activity), and CAC score of the sample were computed for the whole population, and by DASH diet score category. Between-group differences were evaluated using Pearson’s chi-square for categorical variables. For continuous variables, tests of linear trends across groups were assessed using analysis of variance (ANOVA).

Multinomial logistic regression was used to estimate relative risk ratios for the association between DASH diet score category and incident and prevalent hypertension (blood pressure reading ≥130/85 mmHg) at Exam 2. This method is used to model nominal, unordered outcome variables [19,20]. We also evaluated the association between DASH diet score category and continuous systolic and diastolic blood pressure reading using linear regression. Tests for trends were performed based on continuous DASH diet score category.

Potential confounders were identified a priori from a literature review and using causal diagrams [21]. Identified confounders were included in multivariable adjusted models. Educational attainment and income were highly correlated (*p* < 0.0001). There were 19 participants (3%) missing household income; therefore, we adjusted for educational attainment. Models included age-adjustment and multivariable adjustment. Sequentially adjusted models were adjusted for age, sex, percent of life lived in the U.S., education attainment, smoking status, alcohol intake, physical activity, and ordinal CAC score category (Model 1), with the additional adjustment for CAC score, HDL cholesterol, LDL cholesterol, diabetes, and BMI (Model 2), and daily energy intake (Model 3). CAC score is an established predictor of hypertension by the American College of Cardiology [22,23,24] and may be associated with diet. Therefore, we included CAC as a confounder in our models. We assessed the direct and indirect effects of continuous CAC score on the causal pathway between DASH diet score and hypertension using parametric regression models [25], which were not statistically significant. The association between DASH diet score and blood pressure may differ between men and women, as sex-specific differences have been reported for cardiovascular outcomes [26]. Due to limited sample size, we conducted an exploratory analysis to test the interaction of DASH diet score category and sex using a multiplicative term with an a priori significance *p* value of <0.10. All analyses were performed using STATA 16.1 [27].

## 3. Results

Demographic characteristics of the analytical sample, overall and stratified by DASH diet score categories, are presented in Table 1. The mean age of participants was 55.6 years and approximately half were women. The majority of participants held a bachelor’s degree or higher, reported a household income greater than $100,000 per year, never smoked, did not drink alcohol, and met criteria for “ideal” weekly physical activity. Percent of life lived in the U.S. was inversely related to DASH diet score category. Greater proportions of participants who were women, never smoked, didn’t drink alcohol, or had ideal physical activity were in the highest DASH diet score category.

Table 2 shows the bivariate analyses for DASH diet score category and associated clinical measurements. There was a greater proportion of participants with no hypertension in the highest DASH diet score category, and a greater proportion of those with incident hypertension at Exam 2 were in the lowest DASH diet score category (*p* = 0.01). There was no association between systolic blood pressure and DASH diet score category (*p* = 0.29), but diastolic blood pressure was lower in the highest DASH diet score category (*p* = 0.01). We did not find an association between DASH diet score category and CAC score at baseline (*p* = 0.21).

Multivariable adjusted models are presented in Table 3. The relative risk ratio (RRR) of incident hypertension versus having no hypertension at Exam 2 was 67% lower among participants in the highest DASH diet score category (aRRR: 0.33; 95% CI: 0.13, 0.82; p_trend_ = 0.02) compared with those in the low DASH diet score category in fully adjusted models. Diastolic blood pressure was 3 mmHg lower among participants in the highest DASH diet score category (aβ: −3.32; 95% CI: −5.59, −1.04; p_trend_ = 0.004) compared with those in the low DASH diet score category in the age-adjusted model, but the association was no longer statistically significant in the fully adjusted models.

In exploratory analyses, sex did not modify associations between DASH diet score category and incident (*p* = 0.58) or prevalent hypertension (*p* = 0.31), systolic blood pressure (*p* = 0.81), or diastolic blood pressure (*p* = 0.45). Stratified analyses are presented in Appendix A. We examined the relationship stratified by sex due to previously reported differences in the literature [28]. Incident hypertension was 77% lower among men in the highest DASH diet score category (aRRR: 0.23; 95% CI: 0.06, 0.82; p_trend_ = 0.03) compared with men in the low DASH diet score category in the fully adjusted model (Appendix A). A directional relationship between DASH diet score category and incident hypertension among women was observed, which was not statistically significant. There was no association between DASH diet score category and prevalent hypertension or systolic blood pressure among all participants, or when the sample was stratified by sex (Appendix A).

Appendix A shows the relationship between the DASH diet component scores and hypertension and blood pressure outcomes. Incident hypertension was 74% lower among participants with the highest grain intake compared with those in the lowest grains quintile (aRRR: 0.26; 95% CI: 0.11, 0.62; *p* < 0.0001) in the fully adjusted models. There was no association between the individual DASH diet component scores and prevalent hypertension or blood pressure.

## 4. Discussion

In this cohort of middle- to older-aged South Asian adults in the U.S., high adherence to the DASH diet was associated with lower risk of developing hypertension after 5 years of follow-up. Incident hypertension at Exam 2 was lower among all participants in the highest DASH diet score category and highest grain intake in the fully adjusted models. To our knowledge, the present analyses are the first to assess the relationship between the DASH diet and blood pressure among South Asians.

Research has identified that South Asians have a higher prevalence of metabolic syndrome, greater insulin resistance, and altered metabolic profile when compared with non-Hispanic whites, indicating that there may be biological differences driving higher cardiovascular disease risk [1]. Previous studies using MASALA data hypothesize that South Asians have lower beta cell function [29,30], which may have broader implications for cardiometabolic health, but more research is needed. There are several biological mechanisms through which the DASH diet provides a protective effect on cardiovascular health. The DASH diet is unique in its specific combination of food and nutrient components. In the initial DASH diet trial [31], investigators found that participants following the DASH diet had higher intakes of several micronutrients due to the emphasis on dietary components beyond fruit and vegetable intake only. We found that higher whole grain intake was associated with lower risk for incident hypertension at follow-up. This finding is supported by a 2017 meta-analysis that found that participants who consumed at least 30 g (or about 2 servings) of whole grains per day were at lower risk of hypertension compared with those who did not [32]. This relationship is biologically plausible due to the rich micronutrient and fiber content of whole grains, which have been consistently linked to lower blood pressure [33]. Our result of lower incident hypertension among participants with higher DASH diet score is also consistent with previous clinical trials where participants randomized to a DASH-style diet had lower blood pressure at follow-up compared with those following the control diet [34,35]. Overall, the DASH diet underscores a combination of food group intakes that supports a wide intake of vitamins and minerals that have an antihypertensive effect.

Our finding of an association between high DASH diet adherence and lower incidence of hypertension is in line with previous cohort studies. One prospective cohort of Brazilian adults without hypertension at baseline found that, compared with those with lower adherence, participants with the highest DASH diet adherence had a 26% lower risk of incident hypertension at four-year follow-up (hazard ratio 0.74; 95% CI: 0.57, 0.95) [36]. Similarly, a prospective cohort of Chinese adults found that participants who followed a DASH-style diet had a 15% lower risk of incident hypertension at 11-year follow-up compared with those with the lowest DASH adherence (hazard ratio 0.85; 95% CI: 0.73, 0.98) [37]. Several large prospective cohort studies have established that incident hypertension is lower among participants with high adherence to a DASH-style diet [38,39,40], or dietary intake with DASH components, such as high whole grain intake [41], and fruit and vegetable intake [42]. Notably, previous large cohorts that have failed to find an association between the DASH diet and incident hypertension consist of samples that are predominantly non-Hispanic white participants [43,44,45]. As South Asians are at greater risk of morbidity and mortality related to ASCVD [1], our results, interpreted together with findings from other cohort studies, indicate that a DASH-style diet may benefit population groups that are the most vulnerable to adverse cardiovascular health outcomes.

The present analyses uniquely adjusted models for CAC score, which has not been included in other cohort studies examining incident hypertension. While previous studies may not have collected CAC, the faster CAC progression among South Asian men compared to other race/ethnic groups, and its role in predicting both hypertension and ASCVD, necessitates the inclusion of CAC in investigating the relationship between diet and hypertension [22,23]. Previous studies have established that there is an association between dietary intake and CAC score [46,47,48], but more research is needed to characterize the relationship. One recent review regarding CAC’s ability to predict incident hypertension provides compelling evidence that CAC measurement can optimize the prevention and treatment of hypertension and ASCVD [49].

Our study has some limitations that should be noted. The DASH diet score was calculated using an FFQ, which has established limitations, including imperfect measurement of diet and participant reporting bias [50,51]. The FFQ allows us to examine the impact of overall annual dietary intake, rather than being limited to 24 h dietary recalls or seven-day food records, which have the known limitation of bias based on when diet assessment was conducted, and can often be more labor-intensive to examine the individual diets of participants, rendering the tools impractical in large cohort studies [52]. Use of the SHARE FFQ, validated for measuring the diet of South Asians in Canada [11], helps to mitigate missing dietary data. The Fung et al. scoring method to calculate the DASH diet by quintiles of intake also minimizes misclassification of estimated individual intake [17]. Moreover, use of the Fung et al. method allowed us to tailor the scoring to include traditional South Asian food items, such as daal and raita, ensuring the diet pattern assessment was cultrually relevant to this population group. While previous research has examined the effect of acculturation on dietary patterns [53], subsequent studies should consider the change in diet over time among South Asians who have immigrated to the U.S. Another limitation of the sample is that the cohort is relatively small, with most participants immigrating from India with high educational attainment and family income, limiting the external generalizability of the findings. Recruitment is underway to include a more diverse sample of South Asians in the U.S.

The present study also has important strengths, including its prospective design facilitating the analysis of the incident hypertension finding at five-year follow-up in the first multi-center cohort study of South Asian adults in the U.S. Additionally, while the initial DASH diet trial was designed to investigate a dietary pattern that could be followed by non-vegetarians and include commonly available foods in the U.S. [29], operationalizing the dietary pattern using FFQ data demonstrates that the DASH diet benefits can be applied to vegetarians and participants that consume food items that are consistent with South Asian dietary patterns (such as daal, roti, okra). This uniquely allowed us to maintain traditional South Asian dietary features in our DASH diet score.

## 5. Conclusions

Higher DASH diet adherence was associated with lower incidence of hypertension at five-year follow-up among a cohort of older-aged South Asian adults in the U.S. Results from the current study support the early emphasis of diet-related recommendations as a prevention strategy for hypertension among populations at increased risk for the early development of ASCVD. Future studies should consider specific interventions that emphasize a DASH-style diet inclusive of culturally relevant foods to promote compliance among South Asian adults in the U.S.

## Figures and Tables

**Figure 1 nutrients-15-03611-f001:**
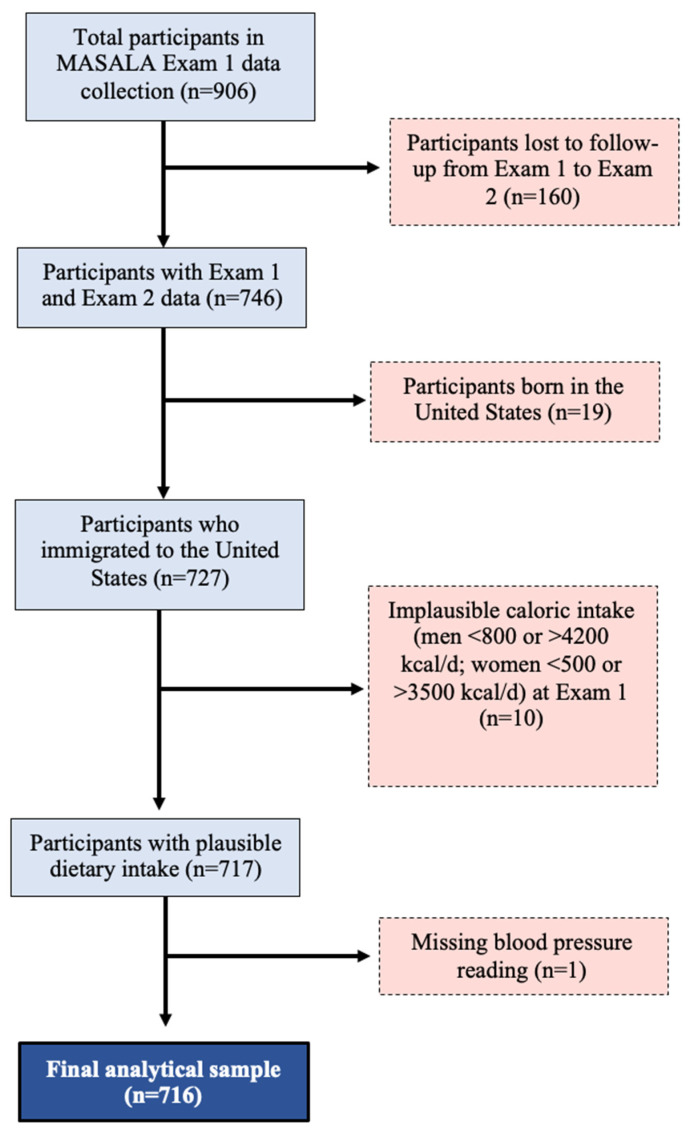
Creation of the analytical sample.

**Table 1 nutrients-15-03611-t001:** Participant characteristics at Exam 1 by DASH diet score category, among South Asian adults in the MASALA Study (*n* = 716).

Characteristic	Total Study Population (*n* = 716)	DASH Score 13–20 (Lowest) (*n* = 127)	DASH Score 21–28 (*n* = 444)	DASH Score 29–35 (Highest) (*n* = 145)	*p* Value
Demographic and Diet Information (Exam 1)					
Age (y) (mean (SD))	55.6 (9.2)	54.2 (9.6)	55.9 (9.2)	55.6 (9.0)	0.18
Women (*n* (%))	322 (45.0)	38 (29.9)	197 (44.4)	87 (60)	<0.001
Percent of life lived in U.S. (mean (SD))	48.6 (16.7)	51.2 (17.4)	48.8 (16.8)	45.8 (15.4)	0.03
Bachelor’s Degree or Higher (*n* (%))	640 (89.4)	108 (85.0)	399 (89.9)	133 (91.7)	0.18
Income > $100 K (*n* (%))	461 (66.4)	85 (67.5)	276 (64.9)	100 (69.9)	0.53
Energy Intake (mean (SD))	1681.9 (492.1)	1631.3 (489.2)	1655.9 (508.5)	1805.9 (421.5)	0.003
Behavioral Risk Factors (Exam 2)					
Never Smoked (*n* (%))	596 (83.2)	92 (72.4)	369 (83.1)	135 (93.1)	<0.001
No Alcohol Intake (*n* (%))	471 (65.8)	55 (43.3)	304 (68.5)	112 (77.2)	<0.001
Physical Activity * (*n* (%))					0.005
Poor	102 (14.3)	22 (17.3)	67 (15.1)	13 (9.0)	
Intermediate	140 (19.6)	35 (27.6)	84 (18.9)	21 (14.5)	
Ideal	474 (66.2)	70 (55.1)	293 (66.0)	111 (76.6)	

* Typical Week’s Activity Survey (Poor indicates no activity; Intermediate indicates 1–149 min of moderate or 1–74 min of vigorous activity per week; Ideal indicates ≥150 min of moderate or ≥75 min of vigorous activity per week; *p* values were estimated by analysis of variance for continuous variables and Pearson’s chi-square for categorical variables.

**Table 2 nutrients-15-03611-t002:** Clinical measurements and medication use by DASH diet score category, among South Asian adults in the MASALA study (*n* = 716).

Characteristic	Total Study Population (*n* = 716)	DASH Score 13–20 (Lowest) (*n* = 127)	DASH Score 21–28 (*n* = 444)	DASH Score 29–35 (Highest) (*n* = 145)	*p* Value
Hypertension and Blood Pressure (Exam 2)					
Incident Hypertension (*n* (%))					0.01
None	362 (50.6)	65 (51.2)	208 (46.9)	89 (61.4)	
Incident (from Exam 1 to Exam 2)	93 (13.0)	20 (15.8)	64 (14.4)	9 (6.2)	
Prevalent/Existing from Exam 1	261 (36.5)	42 (33.1)	172 (38.7)	47 (32.4)	
Systolic Blood Pressure (mmHg) (mean (SD))	127.8 (17.5)	128.6 (15.6)	128.3 (17.5)	125.8 (18.9)	0.29
Diastolic Blood Pressure (mmHg) (mean (SD))	75.1 (9.6)	77.0 (9.2)	75.0 (9.7)	73.6 (9.7)	0.01
Clinical Measurements (Exam 1)					
CAC Score Category (Exam 1) (*n* (%))					0.21
0	414 (58.0)	68 (53.5)	253 (57.1)	93 (64.6)	
1–100	37 (5.2)	9 (7.1)	19 (4.3)	9 (6.3)	
101–400	123 (17.2)	19 (15.0)	84 (19.0)	20 (13.9)	
>400	140 (19.6)	31 (24.4)	87 (19.6)	22 (15.3)	
HDL Cholesterol (mean (SD))	49.9 (13.1)	48.5 (12.6)	50.1 (13.3)	50.4 (12.8)	0.44
LDL Cholesterol (mean (SD))	111.3 (32.0)	112.5 (28.8)	111.4 (33.7)	110.0 (29.1)	0.81
BMI (kg/m^2^) (mean (SD))	25.8 (3.9)	25.8 (3.8)	25.9 (4.0)	25.5 (3.6)	0.57
Diabetes (*n* (%))	171 (23.9)	31 (24.4)	111 (25.0)	29 (20.0)	0.33

**Table 3 nutrients-15-03611-t003:** Sequentially adjusted models of hypertension and blood pressure by DASH diet score category, among South Asian adults in the MASALA study (*n* = 716).

	DASH Score 13–20 (Lowest) (*n* = 127)	DASH Score 21–28 (*n* = 444)		DASH Score 29–35 (Highest) (*n* = 145)		P_trend_ *
	Reference	RRR/β (SE)	95% CI	RRR/β (SE)	95% CI	
Incident Hypertension ^1^						
Unadjusted	1.00	1.00 (0.29)	0.56, 1.78	0.33 (0.14)	0.14, 0.77	0.01
Age Adjusted	1.00	0.92 (0.27)	0.51, 1.65	0.29 (0.13)	0.12, 0.70	0.01
Model 1 ^+^	1.00	0.97 (0.31)	0.53, 1.80	0.32 (0.15)	0.13, 0.79	0.01
Model 2 ^++^	1.00	1.00 (0.32)	0.53, 1.88	0.33 (0.16)	0.13, 0.84	0.02
Model 3 ^+++^	1.00	1.00 (0.32)	0.53, 1.89	0.33 (0.16)	0.13, 0.85	0.02
Prevalent Hypertension ^1^						
Unadjusted	1.00	1.28 (0.29)	0.83, 1.98	0.82 (0.22)	0.48, 1.38	0.34
Age Adjusted	1.00	1.12 (0.27)	0.70, 1.80	0.68 (0.20)	0.38, 1.20	0.14
Model 1 ^+^	1.00	1.28 (0.33)	0.77, 2.11	0.85 (0.27)	0.46, 1.57	0.47
Model 2 ^++^	1.00	1.35 (0.38)	0.77, 2.36	0.97 (0.34)	0.49, 1.93	0.79
Model 3 ^+++^	1.00	1.36 (0.39)	0.77, 2.37	1.01 (0.36)	0.50, 2.02	0.89
Systolic Blood Pressure						
Unadjusted	0.00	−0.30 (1.76)	−3.75, 3.16	−2.77 (2.12)	−6.94, 1.40	0.18
Age Adjusted	0.00	−1.36 (1.67)	−4.63, 1.91	−3.67 (2.01)	−7.62, 0.27	0.06
Model 1 ^+^	0.00	−0.58 (1.72)	−3.96, 2.80	−2.45 (2.13)	−6.63, 1.74	0.24
Model 2 ^++^	0.00	−0.54 (1.72)	−3.92, 2.84	−1.87 (2.14)	−6.08, 2.34	0.37
Model 3 ^+++^	0.00	−0.58 (1.73)	−3.97, 2.81	−2.04 (2.18)	−6.32, 2.23	0.34
Diastolic Blood Pressure						
Unadjusted	0.00	−2.02 (0.96)	−3.91, −0.12	−3.47 (1.16)	−5.76, −1.18	0.003
Age Adjusted	0.00	−1.83 (0.96)	−3.72, 0.05	−3.32 (1.16)	−5.59, −1.04	0.004
Model 1 ^+^	0.00	−0.81 (0.97)	−2.72, 1.09	−1.51 (1.20)	−3.87, 0.86	0.21
Model 2 ^++^	0.00	−0.87 (0.98)	−2.79, 1.04	−1.17 (1.21)	−3.55, 1.22	0.35
Model 3 ^+++^	0.00	−0.87 (0.98)	−2.79, 1.04	−1.17 (1.23)	−3.59, 1.25	0.35

^1^ Incident and prevalent hypertension at Exam 2, defined by NCEP criterion (≥130/85 mmHg); RRR: Relative Risk Ratio for the multinomial logistic regression model; SE: Standard Error; CI: Confidence Interval; * *p*-trend calculated by unadjusted linear regression, using three groups of DASH diet score as an ordinal variable; ^+^ Model 1: Adjusted for age, sex (male/female), percent life lived in the U.S., education (≥Bachelors/<Bachelors), physical activity (ideal, intermediate, poor), smoking (current/former vs. never), alcohol intake (no consumption/any); ^++^ Model 2: Model 1 + CAC score category (0, 1–100, 101–400, >400), HDL cholesterol, LDL cholesterol, diabetes, BMI (kg/m2); ^+++^ Model 3: Model 2 + energy (kcal/d).

## Data Availability

Data described in the article, code book, and analytic code will not be made publicly available. Further information including the procedures to obtain and access data from the MASALA study is described at https://www.masalastudy.org/for-researchers (accessed on 14 August 2023).

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
