# Peer review of "Concordance between Dash Diet and Hypertension: Results from the Mediators of Atherosclerosis in South Asians Living in America (MASALA) Study"

_nutrients, 2023, doi:10.3390/nu15163611_

Round 1
Reviewer 1 Report
Non-pharmacological treatment, including diet therapy, is the basis for the treatment of both arterial hypertension and a number of atherosclerotic diseases. I have a few remarks, the clarification of which will improve the meaning of the work:
- the most important - the problem of physical activity. You divide the patients into three groups. In addition, you define optimal activity as 75 minutes of vigorous activity per week. After all, it's less than 11 minutes a day, it doesn't have any metabolic effect, it doesn't work. In this way, it may turn out that this activity was bad for everyone, and the calculations do not reflect reality. Surely you have data that will allow you to transform these 3 degrees into a numerical value, e.g. from 0 to 10, which will allow you to reflect the activity in a linear way.
- there is no data on the type of activity - people were not very active, see above. However, this is a group of quite young and quite slim people who can be active. run, take part in triathlons, but also do more "down-to-earth" sports such as cycling, swimming, Nordic walking, rowing (both on the water and on the ergometer), etc. Maybe they spend a few hours a day on these activities ... Please be clear at work distinguish the percentage of very active people and use a different division of activity, taking into account aerobic and strenuous activities.
- whether the blood pressure holter was recorded during physical activity. How did blood pressure behave during exercise.
- were the examined persons undergoing an echocardiographic examination? were there indicators of the influence of activity on the echo (there are differences in aging athlete's heart and echocardiographic evaluation can differentiate between Competitive Sprint- versus Endurance-Trained Master Athletes), moreover, were there indicators of hypertensive heart damage?
Reviewer 2 Report
This is a very interesting paper as it is the first study to examine the relationship between the DASH diet and blood pressure in a South Asian population in U.S. But some modification should be needed to publish in Nutrients.
Supplementary table1: Q3(Medan serving/day) is also important information. They should be added to the table.
Line 51-53: Period of taking DASH diet described in the reference should be wrote to compare the results written in the manuscript.
Table3 Unadjusted model should be shown.
In the Supplementary file5, Q5 indicates that the whole grain intake is 3.5 serves/day. Is this amount more or less than in the literature cited in Lines 249-251? This might be critical point and should be added in the text.
Looking at Supplementary 4, it appears that between Q1 and Q5, there are also contributions from sodium, vegetables, and sugar-sweetened beverages rather than whole grain. Discuusion about these should be added.
The efficacy of the DASH diet has been reported in many cases, but the significance of its effectiveness in South Asians needs to be discussed more detail. It could be that they are more prone to hypertension in terms of molecular mechanisms, or that they are more prone to hypertension in terms of epidemiology.
Reviewer 3 Report
This manuscript describes hypertension incidence according to a quantified metric of adherence to the DASH diet in a South Asian population. 93 cases of incident hypertension were identified over follow-up. Hypertension incidence was almost 70% lower in the highest compared to the lowest category of DASH dietary adherence. There are some areas of the manuscript that merit attention:
1) the main problem with the DASH diet has been replicating this feeding study in the free-living population. Studies on this have uniformly shown that this tremendously effective diet is very difficult to get free-living people to adhere to. This is not mentioned.
2) there is no discussion of the cultural appropriateness of the DASH diet, or components of the DASH diet, for a South Asian population.
3) table 1 shows that there are significant differences between those in the highest and lowest DASH diet score categories. For example shorter duration in the US, women, higher energy intake, never smoking, non-drinkers and those with high levels of physical activity were all more likely to be in the highest DASH diet score category. A tremendous opportunity has been missed to comment on this in the text
4) there are several concerns about how the analyses were undertaken: a) please discuss the rationale for categorizing the DASH diet score - providing a distribution of these scores would help better understand the data, b) how was death and missing data handled in these analyses, and c) given that there were only 93 incident cases, that means that if 10 cases are needed per covariate included, that some of the models reported are arguably over-adjusted
5) the discussion is too long and can be shortened by at least one-third
MINOR
1) there is no need to repeat verbatim numbers in the text that are contained in table 1
2) the discussion of the DASH diet and how it lowers BP is interesting but deficient and also detracts away from the main points of this paper. The deficiency is that the importance of dietary potassium and nitrate to lowering BP in those adhering to the DASH diet is not discussed. In the contest of South Asians the important issues are is this a culturally appropriate diet (or certain components) and if it is, why are so few adhering to it.
3) another limitation of this study is the relatively small sample size prevents precision in the reported risk estimates and limits the ability to undertake stratified analyses
Round 2
Reviewer 1 Report
Thank you for the comprehensive answers. I agree that the assessment of physical activity is consistent with the ACC/AHA, but the levels of activity proposed in these recommendations poorly reflect the actual activity, especially in younger patients, and furthermore, the assessment is hampered by the lack of specifying the type of effort. However, I understand that the authors do not have the exact data and therefore cannot enter them into the manuscript.
Author Response
Thank you for understanding the limitations of the current data set. We appreciate your comments as they help inform future research inquiries.
Reviewer 2 Report
The items I pointed out are well corrected.
Author Response
Thank you very much!
Reviewer 3 Report
The manuscript is improved and there has been substantive attention paid to issues previously raised. However, several issues remain:
MAJOR
1) the issue of the cultural relevance of the DASH diet for South Asians was not really directly addressed - what was highlighted was how the instrument calculating dietary components of the DASH diet included items from the south Asian diet. There is an interesting observation that the DASH diet is less adhered to the longer one stays in the USA - does this vary by sex? The reader has no insight into what aspects of the DASH diet adherence worsen over time - such information would provide keener insight into the reported observations.
2) the finding that the DASH diet reduces hypertension risk in this high-risk population is new but not unexpected. Thus, providing more insight into the findings will further enhance the attractiveness of the paper. For example, is there insight into why women are much more likely to have high adherence to the DASH diet than men in this cohort?
3) I am very leery of the interpretation of the data that the DASH diet did not lower hypertension incidence in women. First, there are fewer than 100 total cases so when you stratify the data by sex and re-run the analyses you are working with a much smaller number of cases. Second, the point estimates, though visibly less impressive in women, are still in the same direction as the point estimates for men. Third, it is hard to biologically explain why this diet would work in men but not women.
MINOR
1) there is no need to display more than one decimal point when reporting blood pressure change
